# Instability of $U_3Si_2$ in pressurized water media at elevated temperatures

Artaches Migdisov [1✉], Haylea Nisbet[1], Nan Li [2], Joshua White[3], Hongwu Xu[1], Andrew Nelson[4] & Robert Roback[1]

Following the Fukushima Daiichi accident, significant efforts from industry and the scientific community have been directed towards the development of alternative nuclear reactor fuels with enhanced accident tolerance. Among the proposed materials for such fuels is a uranium silicide compound ($U_3Si_2$), which has been selected for its enhanced thermal conductivity and high density of uranium compared to the reference commercial light water reactor (LWR) nuclear fuel, uranium oxide ($UO_2$). To be a viable candidate LWR fuel, however, $U_3Si_2$ must also demonstrate that, in the event of this fuel coming in contact with aqueous media, it will not degrade rapidly. In this contribution, we report the results of experiments investigating the stability of $U_3Si_2$ in pressurized water at elevated temperatures and identify the mechanisms that control the interaction of $U_3Si_2$ under these conditions. Our data indicate that the stability of this material is primarily controlled by the formation of a layer of $USiO_4$ (the mineral, coffinite) at the surface of $U_3Si_2$. The results also show that these layers are destabilized at $T > 300\,°C$, leading to the complete decomposition of $U_3Si_2$ and its pulverization due to its full oxidation to $UO_2$.

[1] Los Alamos National Laboratory, Earth and Environmental Sciences Division, Los Alamos, NM, USA. [2] Los Alamos National Laboratory, Center for Integrated Nanotechnologies, Los Alamos, NM, USA. [3] Los Alamos National Laboratory, Materials Science and Technology Division, Los Alamos, NM, USA. [4] Oak Ridge National Laboratory, Nuclear Fuel Development Section, Oak Ridge, TN, USA. ✉email: artas@lanl.gov

The interaction of nuclear fuel with a water coolant is a critical factor in the development of new fuels for light water reactor (LWR) applications. Indeed, in the event of a cladding breach, the interaction of the fuel material with the water-based coolant is inevitable, and can potentially lead to a range of undesirable events, such as the dissolution of U in the coolant, physical pulverization, and "wash-out" of the fuel, and, in an extreme event, contamination of the primary loop with fuel material and fission products, and disruption to the cladding geometry that is ultimately responsible for retaining the thermal hydraulic performance of the core[1,2]. Therefore, in addition to assessing its performance under normal operating conditions, an evaluation of an LWR fuel candidate requires an investigation of its behavior in contact with pressurized water at temperatures and pressures typical of the primary loop[3]. In a previous publication, we reported the first results of experiments performed with $U_3Si_2$ in hydrothermal solutions of controlled redox chemistry at 250–350 °C[4]. This study demonstrated that the stability of this material is highly dependent on the redox conditions in the system. It was shown that while $U_3Si_2$ remained stable for a reasonably long time (30 days) at 300 °C, it quickly (in fewer than 50 h) decomposed at 350 °C, and was pulverized into finely dispersed U oxides. Several hypotheses were proposed in an attempt to explain such contrast in behavior of $U_3Si_2$ at different temperatures: from the formation of protective layers of U oxides[5,6], stabilizing the material at $T \leq 300$ °C, to the hydriding of $U_3Si_2$ associated with the distortion of its matrix and pulverization of the material[7]. These hypotheses, however, were unable to explain several aspects of the observed effects. First, the pulverization effect suggests that the oxidation of $U_3Si_2$ to $UO_2$ likely occurs with an extreme volumetric effect, and therefore, it is unlikely that such a process can lead to the formation of a dense water-impermeable protective layer at the surface of U silicide. This has been indirectly confirmed by SEM images of post-experimental pellets, which suggest high porosity in the layers of U oxides formed[4]. In order to provide efficient protection of the bulk material, the formed layer must be highly dense and impermeable: high-temperature drop-calorimetric measurements of the standard enthalpy of formation of $U_3Si_2$ yielded a value of $-33.2 \pm 3.1$ kJ/mol·at.%[8,9], which suggests that even at room temperature, any contact with water will result in the immediate oxidation of $U_3Si_2$ to U oxides. Second, the hydriding of $U_3Si_2$ does not explain the high stability of this material at temperatures $\leq 300$ °C and near-immediate decomposition at temperatures exceeding this limit. It is unlikely that hydriding occurs as a step-function process that would manifest such drastic effects over such a narrow temperature interval. A noteworthy discovery in our previous publication was the identification of a layer of Si-enriched phase located between the porous crust of $UO_2$ and the unaltered bulk $U_3Si_2$ on post-experimental samples, determined by SEM analyses. A recent study also suggests the formation of a Si-rich phase at certain stages of $U_3Si_2$ oxidation[10]. It is tempting to theorize that this unidentified Si-enriched phase is the component controlling the stability of $U_3Si_2$ in water-dominated systems at $\leq 300$ °C.

Here we demonstrate that the oxidative stability of pure $U_3Si_2$ in pressurized water media is primarily controlled by the formation of a layer of $USiO_4$ (the mineral, coffinite) at the surface of $U_3Si_2$. Our data also suggest that these layers are destabilized at $T > 300$ °C, leading to the complete decomposition of $U_3Si_2$ and its pulverization due to its full oxidation to $UO_2$.

## Results

In this contribution, we have focused on the identification of this unknown phase, and, in doing so, on the experimental refinement of the oxidative behavior of $U_3Si_2$ and the physicochemical controls governing its behavior in pressurized water-dominated systems at elevated temperatures. We approached this task by applying two independent experimental methods. Considering the extremely thin nature of the Si-enriched layers, transmission electron microscopy (TEM) techniques were applied. However, the thin and possibly brittle nature of the protective layers formed at the surface of $U_3Si_2$ can pose challenges for TEM specimen preparation and observation. Therefore, in conjunction with an extensive post-experimental phase characterization, we also conducted hydrothermal solubility experiments to determine the concentrations of U that developed in co-existing solutions with the $U_3Si_2$ pellets at elevated temperatures. It is known that the measured concentrations are characteristic of the solubility of the solid in equilibrium with an aqueous phase, governed by its chemical properties and reactivity. The fact that the Si-enriched phase forms a protective layer at the surface of the $U_3Si_2$ bulk material suggests that it is this phase, and not the bulk material, that is in chemical contact with the aqueous solution, and is, thus, responsible for the solubility levels developed in the co-existing solution. When compared with thermodynamic calculations, these solubility levels can be used to identify the unknown phase if the speciation and thermodynamic properties of the species of the metal of interest (U in this case) are precisely known for the experimental conditions applied and if the properties of potential candidate phases have already been determined or evaluated in the literature. The requirement of knowing the aqueous speciation of U at experimental conditions sets some restrictions on the chemical composition of the experimental solution.

To date, the most reliable and accurate high-T thermodynamic calculations are those performed for NaCl-predominant solutions. This limitation is owing to the relative paucity of activity models tuned and experimentally verified at elevated temperatures. One of the most reliable models is that developed for NaCl-dominated solutions (recommended up to $I = 6$ and $T$ up to 600 °C)[11–13]. Uranium speciation in chloride-dominant high-T solutions is best known at acidic and weakly acidic conditions[14,15]; calculations at higher pH are characterized by higher levels of uncertainty. Thus, in order to accurately model the solubility levels determined in solutions co-existing with $U_3Si_2$, the experiments reported here involved the equilibration of $U_3Si_2$ with weakly acidic NaCl-bearing solutions, as opposed to pure water as used in our previous study. To verify the reproducibility of the modeling, the solubility of $U_3Si_2$ was determined in solutions with varying concentrations of NaCl.

**Solubility experiments**. To ensure the stability of $U_3Si_2$, an isothermal series of experiments were performed between 200 and 250 °C (below 300 °C) with increments of 25 °C. Redox conditions were controlled using the Co/CoO solid-state redox buffer[4]. Each of the experimental series was compared with thermodynamic modeling calculations evaluating the solubility of solid phases that are stable under the redox conditions. The data collected on the solubility of $U_3Si_2$, together with the conditions at which the experiments were performed, and the results of thermodynamic calculations are reported in Supplementary Data 1.

Figure 1 illustrates the solubility results of $U_3Si_2$ obtained in the experimental solutions at 200 and 250 °C (blue circles) compared with those modeled theoretically assuming saturation with respect to $UO_2$ (gray squares). As shown in this figure, the theoretical concentrations of U that $UO_2$ can develop in these solutions are ~4 orders of magnitude lower than the concentrations determined in our experiments. Although the formation of amorphous $UO_2$ can potentially boost the concentrations of U in the solution, the scale of this increase is likely insufficient to

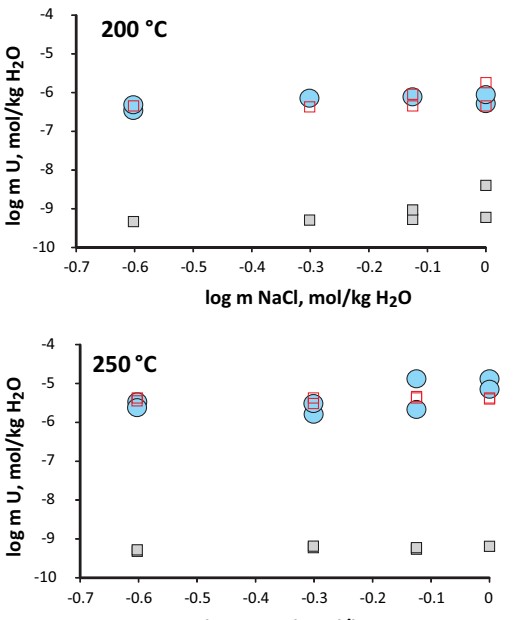

**Fig. 1 The data collected on the solubility of $U_3Si_2$ and the results of thermodynamic calculations.** Concentrations of U determined in solutions co-existing with $U_3Si_2$ at 200 (upper) and 250 °C (lower). Experimental concentrations (blue circles) are compared with the predictions for concentrations that should be developed in equilibrium with $UO_2$ (gray squares) and $USiO_4$ (open red squares). Error bars are smaller than the symbols on the diagram.

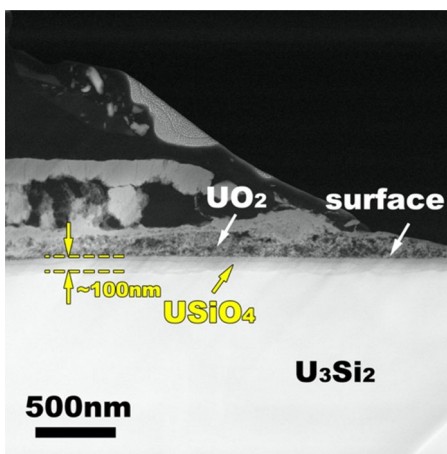

**Fig. 2 Protective layer formed at the surface of $U_3Si_2$.** TEM image of $U_3Si_2$ and the newly formed $USiO_4$ and $UO_2$. Note that the $USiO_4$ layer is dense and thin while $UO_2$ is porous and thick.

achieve the observed effect (the data for amorphous $UO_2$ are available for low temperatures only[16–18]). This disparity suggests that the U-bearing phase interacting with the aqueous solution is not $UO_2$, and is a phase characterized by significantly higher reactivity. As previously mentioned, $U_3Si_2$ (as well as other U silicides) is not stable in contact with water[8,9], and, thus, cannot be considered as a potential candidate. However, the interaction of $U_3Si_2$ with water not only exposes U to oxidation but also Si; it is, therefore, logical to predict the formation of a phase involving both oxidized U and Si. Moreover, the XPS study by Yan et al.[10] suggests the formation of uranium silicates at the interface of $U_3Si_2$ and the aqueous phase, and, thus, of the possible phases that could have coated the surface of $U_3Si_2$ and controlled the solubility of U in the co-existing solutions of our experiments, we considered the mineral coffinite ($USiO_4$) as a likely candidate. Unfortunately, in contrast to $UO_2$[eg.,19], the thermodynamic properties of coffinite are not well defined, and experimental data available in the literature are restricted to values for standard enthalpy and Gibbs free energy of formation[20–22]. Although some first-principles calculations are available for the electronic structure, bonding, and thermodynamic properties of $USiO_4$[23], accurate experimental data for standard entropy and temperature dependence of heat capacity are absent. Thus, several assumptions were made to derive a complete set of thermodynamic properties for this phase[24,25], and these extrapolations were used to provide a rough estimate of the solubility of coffinite at the experimental conditions. Calculated U concentrations based on these values are shown in Fig. 1 (open red squares). Remarkably, these calculated values very closely approximate the solubility values obtained experimentally. This observed similarity further

supports the hypothesis that coffinite is forming a layer at the surface of $U_3Si_2$ and is controlling its stability and oxidative behavior in aqueous media at elevated temperatures.

**TEM studies**. Our TEM studies performed on post-experimental samples confirmed this hypothesis. A thin (<100 nm) and dense layer of $USiO_4$ was identified at the surface of the bulk $U_3Si_2$ (Fig. 2). This layer was covered by a significantly thicker and highly porous layer of $UO_2$. The extreme porosity of the $UO_2$ layer supports the above-mentioned assumption that the oxidation of $U_3Si_2$ to $UO_2$ occurs with a high volumetric effect and, thus, cannot lead to the formation of a wholly protective layer at its surface. The occurrence of this $USiO_4$ layer was further validated through high-resolution TEM imaging and a Fourier transform of the high-resolution area (Fig. 3). The lattice fringes with d-spacings of 2.66 and 1.81 Å are in agreement with the (112) and (321) planes of coffinite, respectively (Fig. 3,- left). The generated electron diffraction pattern is also consistent with the crystallographic symmetry of coffinite (space group $I4_1/amd$) and the angle between (112) and (321) planes is around 81°, which is consistent with the tetragonal coffinite structure (Fig. 3,-right). Although the d-spacing of (112) plane (2.66 Å) of this phase is similar to that of the (210) plane (2.64 Å) of USi, the (321) plane d-spacing (1.81 Å) is quite different from those of other d-spacings of USi; thus, the occurrence of USi is highly unlikely. Moreover, the formation of USi cannot be supported from the viewpoint of phase stability relations: in fact, USi is as unstable with water as $U_3Si_2$[9,26]. It should be noted that the observed orientation of the $USiO_4$/$UO_2$ interface ($USiO_4$ (112) || $UO_2$(020) and $USiO_4$(321) closely parallel to $UO_2$(200)) may not be representative throughout the whole interface.

Noteworthily, although solubility measurement and TEM characterization have their own caveats (discussed above) and more detailed studies of the oxidative behavior of $U_3Si_2$ may be needed, the results from these two different techniques both suggest the formation of a coffinite protective layer at the surface of $U_3Si_2$, giving higher certainty than each study individually.

## Discussion

Coffinite is a mineral known to be difficult to synthesize as it is metastable under a range of conditions and can form only in the presence of amorphous/colloidal silica, which oversaturates the

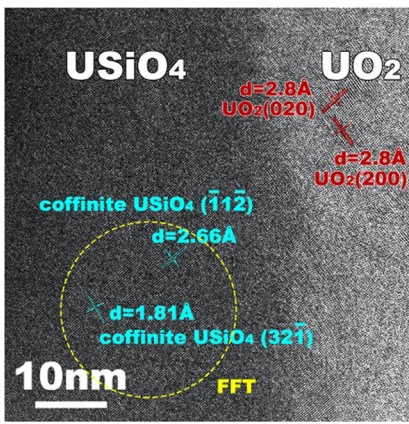 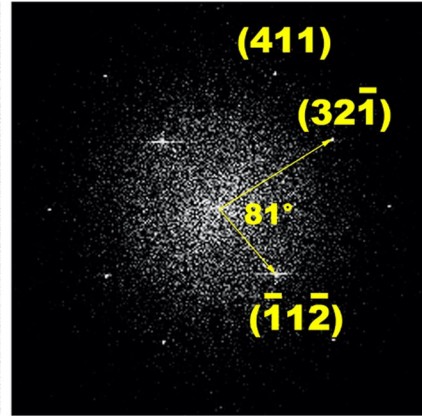

**Fig. 3 Characterization of the protective layer formed at the surface of U₃Si₂.** High-resolution TEM image of the USiO₄--UO₂ interface (left) and electron diffraction pattern of USiO₄ (right).

system with respect to dissolved $SiO_2$[24,27,28]. We, therefore, propose that the oxidation of $U_3Si_2$ to form coffinite occurs in aqueous media as a process involving two stages:

1. Oxidation of $U_3Si_2$ leading to the formation of a porous $UO_2$ layer and the production of amorphous/colloidal silica[6]:

$$U_3Si_2 + 10H_2O = 3UO_2 + 2SiO_2{}^{am} + 10H_2{}^{gas} \quad (1)$$

2. Simultaneous interaction of amorphous/colloidal silica entrapped in the pores with $UO_2$ and $U_3Si_2$ to form coffinite:

$$UO_2 + SiO_2{}^{am} = USiO_4 \quad (2)$$

$$U_3Si_2 + SiO_2{}^{am} + 10H_2O = 3USiO_4 + 10H_2{}^{gas} \quad (3)$$

From these equations, it is shown that the "cementation" reactions (2) and (3) require an excess of silica formed through the initial oxidation of $U_3Si_2$ to $UO_2$ and $SiO_2{}^{am}$ (reaction 1). Based on the microstructures observed in the TEM images, we speculate that oversaturation with respect to $SiO_2$ occurred only in close proximity to the surface of the bulk $U_3Si_2$. At greater distances from the $U_3Si_2$ surface, the concentration of dissolved $SiO_2$ is likely to quickly decrease due to dilution from the surrounding aqueous solution (because of the high porosity of the $UO_2$ crust), where it becomes no longer sufficient to form coffinite. We cannot state with certainty that $USiO_4$ forms fully crystalline layers. Earlier studies[29] demonstrating an amorphous layer rich in U, Si, and O below nanocrystalline $UO_2$ suggest that deviations from rigid stoichiometry are possible within the coffinite layers. However, the close correspondence of the experimentally measured solubilities to those calculated for $USiO_4$ suggests that the portion of the protective layer that is in contact with water should closely correspond to stoichiometric coffinite. It should also be noted that reactions (2) and (3) illustrate a general theorized trend of the surface oxidation of $U_3Si_2$, rather than a precisely determined reaction path. A detailed reaction path of this process will depend on a multitude of factors such as solution chemistry and temperature, and thus clearly requires further experimental investigation. Noticeably, both stages occur with the production of molecular hydrogen and, thus, in parallel can trigger hydriding of $U_3Si_2$, the process that has been theorized in our earlier paper[4] and confirmed in a more recent study[23]. The latter process is known to be significantly destructive to the $U_3Si_2$ structure due to large volumetric effects associated with the formation of hydride forms.

The discovery of a coffinite protective layer on the surface of $U_3Si_2$ also explains the quick and complete pulverization of the $U_3Si_2$ pellets at temperatures above 300 °C. Figure 4 shows the

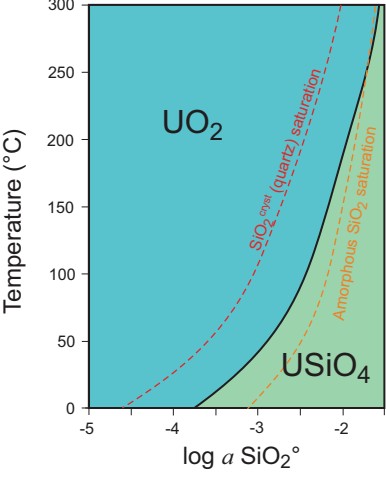

**Fig. 4 Phase stability diagram for UO₂ and USiO₄.** Stability fields of $UO_2$ and $USiO_4$ plotted as a function of temperature and activity of dissolved silica in co-existing solutions. Dashed lines on the diagram represent calculated saturation levels with respect to crystalline (red) and amorphous (orange) $SiO_2$ (reproduced from ref. [25]).

stability field diagram of coffinite and uraninite as a function of temperature and the activity of silica in aqueous solutions (ref. [25]). The diagram also plots the saturation lines of $SiO_2{}^{aq}$ concentrations with respect to quartz and amorphous silica. As is shown in this figure, coffinite is not stable in equilibrium with quartz, which develops concentrations of $SiO_2{}^{aq}$ too low to form $USiO_4$. In contrast, if amorphous silica is present in the system, the concentration of $SiO_2{}^{aq}$ is promoted to levels sufficient to form coffinite at temperatures <300 °C. At higher temperatures, however, the curve corresponding to the saturation of amorphous silica shifts into the stability field of uraninite ($UO_2$), intersecting the stability boundary between these two phases at a temperature of ~300 °C. This suggests that at temperatures above this threshold, the presence of amorphous silica is not sufficient anymore to stabilize coffinite; rather, $UO_2$ is the stable phase. Furthermore, this observation suggests that the protective layer of $USiO_4$ discovered in our experiments on the surface of $U_3Si_2$ simply cannot form at temperatures above 300 °C and that at these temperatures, the oxidation of $U_3Si_2$ occurs immediately and leads to the complete transformation of $U_3Si_2$ to U oxides. This is precisely the situation that we observed in our earlier experiments[4] (i.e. the pulverization of $U_3Si_2$ pellets at >300 °C). It should be noted that this diagram was based on approximate

theoretical calculations of the stability of coffinite at elevated temperature, and the exact temperature at which the line corresponding to the saturation of $SiO_2^{am}$ departs from the stability field of coffinite has not yet been well-established. Nevertheless, even when considering the approximate nature of the diagram, the stability fields of coffinite and $UO_2$ are consistent with our experimental observations and justify the trends in the observed oxidative behavior of $U_3Si_2$ discussed above.

## Conclusions

Our data therefore indicate that the oxidative stability of pure $U_3Si_2$ in pressurized water media is primarily controlled by the formation of a layer of $USiO_4$ (coffinite) at the surface of $U_3Si_2$. The results also show that these layers are destabilized at $T > 300$ °C, leading to the complete decomposition of $U_3Si_2$ and its pulverization due to its full oxidation to $UO_2$. Given that the operating temperatures of a nuclear reactor are above 300 °C, this suggests that the use of pure $U_3Si_2$ as an alternative LWR fuel is questionable because of its instability above 300 °C. A possible approach to mitigate this effect is to dope $U_3Si_2$ with other elements[30,31] to improve the stability of the formed protective layer at elevated temperatures. The results of this study demonstrate that when considering a nuclear fuel candidate in the U-Si system, a similar assessment is required to ensure its stability under hydrothermal conditions.

## Methods

**Material preparation**. $U_3Si_2$ was synthesized via arc melting of the pure metal constituents. Stoichiometric ratios of U and Si were prepared for arc-melting by first removing the oxide layer from the U metal via SiC grinding disks and weighing the appropriate masses of U and Si to ±0.02 mg. The arc melting setup had three welding leads, two of which were used to distribute the current evenly across a 5 g boule, while the third melted an internal Ti getter to remove any oxygen impurities in the chamber. The atmosphere within the arc melter consisted of ultra-high-purity Ar that was passed over a Cu getter to remove oxygen impurities in the gas prior to introduction in the arc melter chamber. The $U_3Si_2$ boule was melted and rotated five times in order to evenly distribute the constituents. The preparation methodology, resulting in phase purity, and microstructure of the $U_3Si_2$ match those presented in the previous investigations[26]. Samples for the solubility experiments were prepared by fracturing a piece from the cast boule using an $Al_2O_3$ mortar and pestle.

## Solubility experiments

*Experimental procedure*. The experiments involved the determination of the solubilities of $U_3Si_2$ in aqueous solutions of various concentrations of NaCl. The experiments were performed at 200, 225, and 250 °C under controlled redox conditions (Co/CoO solid-state redox buffers) in light-weight test tube-sized autoclaves (35–40 cm³ internal volume), manufactured from Titanium Grade 2. Temperatures lower than 200 °C were not investigated due to the poor performance of solid-state redox buffers at these conditions (kinetic hindrance). The upper threshold of 250 °C was selected based on our previous study[4] to remain within the stability field of $U_3Si_2$. The autoclaves were passivated with a layer of $TiO_2$ to ensure its chemical inertness. The experimental techniques employed in this study are similar to those reported in our earlier study[4]; for details not covered in the following description, including discussion of the principles of solid-state buffer application, the readers are referred to the above paper.

Experimental solutions were prepared with de-ionized, $CO_2$-free water and NaCl (Fisher Scientific, A.C.S.) with concentrations ranging from 0.25 to 1.0 m (mol/kg). The solutions were adjusted to a $pH^{25°C}$ of approximately 2 by adding the appropriate amount of HCl (Fisher Scientific, Optima grade). The autoclaves were first loaded with two separate holders (1–5 mm diameter fused quartz or gold tubes; upper end open) containing lumps of $U_3Si_2$ and Co/CoO redox buffer. Next, an aliquot of the experimental solution was added. The volume of the added solution was calculated to ensure the solution did not come in contact with $U_3Si_2$ at ambient conditions but would expand and flush the holder at the experimental temperature due to thermal expansion. This approach ensures that the solubility determined in the experiments corresponds only to the experimental temperature and is not affected by processes that may occur during heating/quenching of the autoclaves. The holders containing solid-state redox buffers were sufficiently long to ensure the experimental solutions did not flood the redox buffers, and $fO_2$ re-equilibration occurred through the gas phase. The autoclaves were purged with high-purity argon gas (Matheson Tri-Gas, Ultrapure) immediately before being capped and sealed using a Grafoil® O-ring. After sealing, the autoclaves were heated to the experimental temperature in a ThermoFisher Scientific Furnace (±0.5 °C) until equilibrium/steady state was attained (see below). After completion of the

experiments, the autoclaves were air-quenched to room temperature and the holders containing the solid phases were removed for subsequent TEM analysis. Post-experimental pH was measured potentiometrically using an Orion glass double-junction electrode and a set of calibration standards with identical NaCl concentrations to the experimental solutions. After, 3–5 ml of concentrated $HNO_3$ (Fisher Scientific, TM grade) was added to each autoclave to dissolve any U that may have precipitated on the inside walls during cooling. Finally, the concentrations of U in the resulting solutions were analyzed by ICP-MS. The data collected on the solubility of $U_3Si_2$, together with the conditions at which the experiments were performed, are reported in Supplementary Data 1.

The time required to attain equilibrium/steady state was determined by a set of 10 experiments with identical solution compositions (NaCl = 0.5 m; pH = 2.0) performed for 1, 2, 3, 5, 7, 9, 10, 11, 13, and 14 days at 200 °C. After ~3–5 days, U concentrations measured in the experimental solutions became constant, suggesting that equilibrium (or steady-state) had been reached. As equilibrium will be kinetically favored at higher temperatures, this time series suggests that the concentrations measured in experiments exceeding 5 days correspond to those of isothermal solubility. All experiments reported in this study were performed for a minimum duration of six days. Similar to the abovementioned experiments, two types of holders were used: fused quartz tubes (1,3, 9, 11, and 14 days) and gold holders (2, 5, 7, 10, and 13 days). Agreement between the results produced by the two types of holders suggests that the material does not affect the processes occurring in the experimental system.

*Thermodynamic calculations*. Thermodynamic modeling calculations were performed to evaluate the solubility of $UO_2$ (U oxide stable at the redox conditions set by Co/CoO redox buffer)[19] and $USiO_4$ (coffinite)[25] and were compared with the experimental solubility data. The calculations were performed using the HCh software, which minimizes the Gibbs free energy of the system[32]. In addition to uraninite ($UO_2$) and coffinite ($USiO_4$), the model also accounted for the potential formation of $UO_3$, $UO_{2.667}$, $UO_{2.33}$, $UO_{2.25}$, and $UO_2(OH)_2$[21] (these phases were ultimately found to be unstable at experimental conditions). The composition of the aqueous solution was modeled with the following species: $H^+$ [33], $OH^-$ [33], $Na^+$ [34], $NaHSiO_3°$ [35], $NaOH°$ [36], $NaCl°$ [35], $SiO_2°$ [37], $HSiO_3^-$ [35], $Cl^-$ [34], $HCl°$ [38], $U^{4+}$ [36,39], $UO^{2+}$ [36,39], $UO_2°$ [36,39], $HUO_2^+$ [36,39], $HUO_3^-$ [36,39], $UCl^4°$ [15], $UO_2^{2+}$ [36,39], $UO_3°$[36,39], $UO_4^{2-}$[36,39], $HUO_4^-$ [36,39], $UO_2OH^+$ [36,39], $UO_2Cl_2°$ [14], and $UO_2Cl^+$ [14]. In terms of individual ion activities, the calculations employed the extended Debye–Huckel model modified for NaCl-dominated solutions[11–13]:

$$\log \gamma_i = -\frac{A \cdot [Z_i]^2 \cdot \sqrt{I}}{1 + B \cdot \mathring{a}° \cdot \sqrt{I}} + \Gamma + b_\gamma I \qquad (4)$$

where $A$ and $B$ are the Debye–Huckel parameters, $Z_i$, $\Gamma$, and $\mathring{a}°$ are the individual molal activity coefficient, the charge, the molarity to molality conversion factor, and the distance of closest approach of an ion $i$, respectively. The effective ionic strength calculated using the molal scale is $I$ and $b_\gamma$ is the extended-term parameter for NaCl-dominated solutions. The results of the calculations are listed in Supplementary Data 1 together with the experimental values. The thermodynamic data used in calculations can also be found in Supplementary Data 1.

**Post-experimental characterization of solid $U_3Si_2$**. Post-experimental samples of $U_3Si_2$ from the solubility experiments were characterized by TEM to identify the microstructure of the phase. Considering that the thickness of the Si-rich layers at the surface of $U_3Si_2$ increases with temperature[4], the sample for the TEM study was taken from the 250 °C experiments. The samples were prepared by first mounting them in epoxy, followed by sequential polishing steps using SiC grinding discs through a 1 μm diamond slurry. Regions of interest were selected for analysis by preparing thin lamellae via a focused ion beam (Helios, FEI). The FEI Titan 80-300 TM with a monochromator, image aberration corrector, and a PHENIX energy dispersive X-ray spectrometer (EDS) detector was employed to characterize the atomic microstructure of the sample.

## Data availability

All data generated or analyzed in this study are included in Supplementary Data 1.

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

## Acknowledgements

Research presented in this article was supported by the laboratory directed research and development program of Los Alamos National Laboratory under project number 20180007DR. TEM studies were performed at the Center for Integrated Nanotechnologies, an office of science user facility operated by the U.S. Department of Energy (DOE) Office of Science. Los Alamos National Laboratory, an affirmative action equal opportunity employer, is managed by Triad National Security, LLC for the U.S. Department of Energy's NNSA, under contract 89233218CNA000001.

## Author contributions

A.M. conducted the solubility experiments, and developed the dissolution model, H.N. conducted the solubility experiments, N.L. conducted the TEM studies, J.W. synthesized the samples and performed XRD characterization, A.M., H.N., N.L., J.W., H.X., A.N., and R.R. participated in discussions, interpretation of the data, and writing of the manuscript.

## Competing interests

The authors declare no competing interests.
