## [Peer Review File · Communications Chemistry]

Reviewers' comments:

Reviewer #1 (Remarks to the Author):

Very informative and nicely written paper, addressing the issue of stability of a prospective type of nuclear fuel, U_3Si_2 , in contact with water. It is surely very important work interesting for a broad community. From pre-existing research it was apparent that this material is quite stable in contact with water, implying some kind of protection coating. Using TEM, it was directly observed here that the protecting material is coffinite, $USiO_4$. This allowed to reveal why the stability does not extend above 300 deg. C. In this range UO_2 becomes more stable than coffinite, and the protective coating is gone, UO_2 peels off easily and U_3Si_2 desintegrate.

The idea that a silicate is formed at U_3Si_2 oxidation is not entirely new. Using XPS, it was indicated at certain stage of oxidation by Tingwen Yan et al., *J. Nucl. Mater.* 520, 1 (2019). That work also illustrates an evident fact that water is never in direct contact with U_3Si_2 , but an overlayer is formed even at UHV conditions at moderate O_2 exposures. Such overlayer has to exit prior to exposure of U_3Si_2 to an aqueous medium. The reactions 1 and 2, describing a possible reaction route of U_3Si_2 with H_2O at low temperatures, may not be relevant. I suggest that the article is accepted if authors consider in the text this fact.

I have few minor critical points about the context of the work. As coffinite became the key player, it may be also good to mention bonding properties and electronic structure, presented recently in Si Lu et al, *AIP Advances* 10, 075018 (2020). Although authors believe that the role of hydrogen gas in desintegration at higher temperatures is not decisive, it may be good to cite the original work showing that U_3Si_2 absorbs large amount of H in atmosphere of few kPa only (probably similar to pure uranium - S. Maskova et al., *J. Nucl. Mater.* 487, 418 (2017)), expanding the lattice tremendously and breaking the material.

Reviewer #2 (Remarks to the Author):

The main focus of the study was to report the formation of $USiO_4$ layer at the surface of U_3Si_2 when it is exposed in pressurized water. U_3Si_2 fuel has been considered as a potential accident tolerant fuel for pressurized light water reactors, therefore, the results about formation of the new oxide layer and its stability analysis are useful to understand the performance and safety of U_3Si_2 fuel under operating and accident conditions.

The experimental results are well organized and the related analyses are reasonable technically. The conclusion provides a useful recommendation such that assessment on stability of oxide layer is required to ensure the use of U-Si fuel system in LWRs.

Based on the two experimental results of TEM investigation and hydrothermal solubility test, the authors concluded that the oxide layer is a crystalline phase of $USiO_4$. Both experimental results strongly suggest that the oxide layer formed on the surface of U_3Si_2 is likely to be the crystalline phase of $USiO_4$, but I think further experimental data and analysis evaluation are necessary to support the conclusion that the oxide layer is the single crystal phase of $USiO_4$. The revision to address the following questions will improve the clarity of the nature of oxide phase formed at surface for reader's understanding.

1. In Fig. 1, experimental concentrations of U in solution are compared with the predictions for UO_2

and USiO₄. Could you provide additionally the prediction of U concentration in solution for the case that the amorphous or crystalline mixture of UO₂-SiO₂ layer is in equilibrium?

2. TEM analysis was performed to identify the oxide layer and a TEM image and an electron diffraction pattern were presented in Fig. 3 to confirm the formation of USiO₄ layer. Coincidentally, however, the (112) plane spacing (2.66 angstrom) denoted in Fig. 3 is quite similar to that of (210) plane (2.64 angstrom) of USi. Isn't there any possibility that the observed crystalline phase has a different composition than USiO₄? EDS elemental mapping might be useful to clarify the formation of USiO₄.

I would recommend expanding the reference list in relation to the recent progress on the understanding of the oxidation mechanism of U₃Si₂. (for example, <https://doi.org/10.1016/j.corsci.2020.108822>, <https://doi.org/10.1016/j.jnucmat.2020.152517>)

Reviewer #3 (Remarks to the Author):

U₃Si₂ has been considered as a candidate of accident tolerant fuels. Authors studied the stability of U₃Si₂ in pressurized water and the underlying mechanisms. It is an important work for nuclear fuel community to mechanistically understand the corrosion behavior of U₃Si₂. Here are some suggestions and comments authors may address.

1. Does the USiO₄ crystallize well in the whole layer? As shown in the earlier research by Harp et al. in Top Fuel 2016, there is an amorphous layer rich in U, Si and O below nanocrystalline UO₂ layer.
2. Any crystallographic relationships of USiO₄/U₃Si₂ and USiO₄/UO₂?
3. Any amorphous SiO₂ found in corrosion products? Any SiO₂ going to water? In terms of mass balance, could you roughly calculate the thickness ratio of USiO₄ and UO₂ layers if they are dense.
4. Is UO₂ stoichiometric? In J Am Ceram Soc. 2018;101:1004–1008, hyper-stoichiometric UO₂ formed after corrosion of U₃Si₂ under a similar condition. Uranium oxide and silicon oxide formed interesting core-shell structure. Could you please explain why USiO₄ rather than mixed oxides formed in this work?

Reviewers' comments:

Reviewer #1:

The idea that a silicate is formed at U₃Si₂ oxidation is not entirely new. Using XPS, it was indicated at certain stage of oxidation by Tingwen Yan et al., J. Nucl. Mater. 520, 1 (2019).

References and short discussion have been added to the manuscript; please see lines 66 and 147.

The reactions 1 and 2, describing a possible reaction route of U₃Si₂ with H₂O at low temperatures, may not be relevant. I suggest that the article is accepted if authors consider in the text this fact. I have few minor critical points about the context of the work.

The following statement has been added to the manuscript: "It should also be noted that reactions (2) and (3) illustrate a general theorized trend of the surface oxidation of U₃Si₂, rather than a precisely determined reaction path. A detailed reaction path of this process will depend on a multitude of factors such as solution chemistry and temperature, and thus clearly requires further experimental investigation." (lines 323-327)

As coffinite became the key player, it may be also good to mention bonding properties and electronic structure, presented recently in Si Lu et al, AIP Advances 10, 075018 (2020).

Mentioning of this study has been added to the manuscript (lines 153-155).

Although authors believe that the role of hydrogen gas in desintegration at higher temperatures is not decisive, it may be good to cite the original work showing that U₃Si₂ absorbs large amount of H in atmosphere of few kPa only (probably similar to pure uranium - S. Maskova et al., J. Nucl. Mater. 487, 418 (2017)), expanding the lattice tremendously and breaking the material.

As a matter of fact, we did not intend to question the importance of hydriding and its destructive effects on the material integrity. These processes clearly control the stability and integrity of U₃Si₂ pellets at the conditions when the formed protective layers of USiO₄ remain intact and impermeable. Our study is focused on the stability and nature of this protective layer, rather than on the processes occurring in the material's matrix. In order to clarify that we do not downplay the importance of hydriding, we have added a short discussion at lines 327-331.

Reviewer #2:

1. In Fig. 1, experimental concentrations of U in solution are compared with the predictions for UO₂ and USiO₄. Could you provide additionally the prediction of U concentration in solution for the case that the amorphous or crystalline mixture of UO₂-SiO₂ layer is in equilibrium?

The data on the stability of amorphous UO₂ are not available for the temperatures at which our experiments were performed. We therefore can only roughly and very approximately estimate a potential shift in solubilities that could be caused by the presence of this phase. This was mentioned at lines 125-131 of the revised manuscript. The presence of amorphous

SiO₂ does not have any impact on the solubility of U (besides triggering the formation of USiO₄).

2. TEM analysis was performed to identify the oxide layer and a TEM image and an electron diffraction pattern were presented in Fig. 3 to confirm the formation of USiO₄ layer. Coincidentally, however, the (112) plane spacing (2.66 angstrom) denoted in Fig. 3 is quite similar to that of (210) plane (2.64 angstrom) of USi. Isn't there any possibility that the observed crystalline phase has a different composition than USiO₄? EDS elemental mapping might be useful to clarify the formation of USiO₄.

Unfortunately, we were not successful in collecting the EDS data for the sample. However, we have re-visited and re-indexed the available TEM scans and provided additional information on Figure 3. In addition to the (112) plane spacing, we have also added the (321) plane spacing (1.81 Å) and the angle between (112) and (321) planes. All of these values match the USiO₄ structure, permitting a reliable identification of this phase. In addition to the Figure, we also discuss this at lines 201-209 of the revised manuscript.

I would recommend expanding the reference list in relation to the recent progress on the understanding of the oxidation mechanism of U₃Si₂. (for example, <https://doi.org/10.1016/j.corsci.2020.108822>, <https://doi.org/10.1016/j.jnucmat.2020.152517>)

References have been added to the manuscript.

Reviewer #3:

1. Does the USiO₄ crystallize well in the whole layer? As shown in the earlier research by Harp et al. in Top Fuel 2016, there is an amorphous layer rich in U, Si and O below nanocrystalline UO₂ layer.

At lines 318-323 the following statement was added to the manuscript “We cannot state with certainty that USiO₄ forms fully crystalline layers. Earlier studies²⁹ demonstrating an amorphous layer rich in U, Si and O below nanocrystalline UO₂ suggest that deviations from rigid stoichiometry are possible within the coffinite layers. However, close correspondence of the experimentally measured solubilities to those calculated for USiO₄ suggests that the portion of the protective layer that is in contact with water should closely correspond to stoichiometric coffinite.”

2. Any crystallographic relationships of USiO₄/U₃Si₂ and USiO₄/UO₂?

Information on the relationship between USiO₄ and UO₂ has been added to Figure 3 and is discussed at lines 205-209 of the revised manuscript.

3. Any amorphous SiO₂ found in corrosion products? Any SiO₂ going to water? In terms of mass balance, could you roughly calculate the thickness ratio of USiO₄ and UO₂ layers if they are dense.

Unfortunately, our experimental technique did not permit the determination of SiO₂ in aqueous solutions (most of the experiments were performed in quartz tubes and the solution

was therefore saturated with respect to SiO_2). Considering the very high porosity of UO_2 layers, calculation of the thickness ratio is highly unreliable.

4. *Is UO_2 stoichiometric? In J Am Ceram Soc. 2018;101:1004–1008, hyper-stoichiometric UO_2 formed after corrosion of U_3Si_2 under a similar condition. Uranium oxide and silicon oxide formed interesting core-shell structure. Could you please explain why USiO_4 rather than mixed oxides formed in this work?*

Based on the data collected in our study we cannot conclude on how stoichiometric UO_2 is. We explain the formation of USiO_4 rather than UO_2 by oversaturation of the system with $\text{SiO}_{2,\text{aq}}$ due to the formation of the amorphous silica (which boosts the concentration of SiO_2 in the porous solution). This is discussed at lines 333-339 and is illustrated in Figure 4.

REVIEWERS' COMMENTS:

Reviewer #2 (Remarks to the Author):

The authors successfully addressed the issues and revised the manuscript appropriately. I think the revised manuscript is acceptable for the publication in the journal.

Reviewer #3 (Remarks to the Author):

The authors have address the comments.